# New Insights into Microbial Degradation of Cyanobacterial Organic Matter Using a Fractionation Procedure

**DOI:** 10.3390/ijerph19126981

**Published:** 2022-06-07

**Authors:** Jing Chen, Yongqiang Zhou, Yunlin Zhang

**Affiliations:** 1Nanjing Institute of Geography and Limnology, Chinese Academy of Sciences, Nanjing 210008, China; chenj@niglas.ac.cn (J.C.); ylzhang@niglas.ac.cn (Y.Z.); 2University of Chinese Academy of Sciences, Beijing 100049, China

**Keywords:** cyanobacterial blooms, dissolved organic matter, nutrient, bacterial abundance, carryover effects

## Abstract

Cyanobacterial blooms caused by phytoplankton *Microcystis* have occurred successively since 1980 in Lake Taihu, China, which has led to difficulty collecting clean drinking water. The effects of cyanobacterial scum-derived dissolved organic matter (DOM) on microbial population variations and of algal-derived filtrate and algal residual exudative organic matter caused by the fraction procedure on nutrient mineralization are unclear. This study revealed the microbial-regulated transformation of DOM from a high-molecular-weight labile to a low-molecular-weight recalcitrant, which was characterized by three obvious stages. The bioavailability of DOM derived from cyanobacterial scum by lake microbes was investigated during 80-d dark degradation. Carbon substrates provided distinct growth strategy links to the free-living bacteria abundance variation, and this process was coupled with the regeneration of different forms of inorganic nutrients. The carryover effects of *Microcystis* cyanobacteria blooms can exist for a long time. We also found the transformation of different biological availability of DOM derived from two different cyanobacterial DOM fractions, which all coupled with the regeneration of different forms of inorganic nutrients. Our study provides new insights into the microbial degradation of cyanobacterial organic matter using a fractionation procedure, which suggests that the exudate and lysate from degradation products of cyanobacteria biomass have heterogeneous impacts on DOM cycling in aquatic environments.

## 1. Introduction

Clean drinking water plays a critical role in human physical health and welfare [1]. Lakes are the most important clean drinking water sources in China because they can provide a relatively stable water supply with few nature fluctuations (i.e., floods, droughts, and freezing) [2]. Lake Taihu (30°05′–32°08′ N, 119°08′–121°55′ S) is the third largest freshwater lake in China. The lake watershed covers approximately 36,500 km^2^ and is an important drinking water source for more than 10 million people and several modern cities, such as Shanghai, Wuxi, and Suzhou. Moreover, Lake Taihu also serves numerous industrial, agricultural, and municipal activities [3]. However, harmful cyanobacteria blooms caused by the phytoplankton *Microcystis* have occurred successively since 1980 in Lake Taihu, which have directly threatened the safety of drinking water and triggered serious health and social problems, particularly the “Water supply crisis” in Wuxi in 2007 [4,5,6,7]. Qin et al. [8] promoted a large-scale integrated monitoring and forecasting system for cyanobacterial bloom management in Lake Taihu, and they found that the distribution area of cyanobacterial blooms reached 997.5 km^2^ in 2011, which was the maximum value for the four consecutive years of monitoring (2009–2012). Generally, cyanobacterial bloom growth can be influenced by temperature, daylight, water turbulence, pH, and macronutrient. In addition, wind-induced hydrodynamic effects, such as sediment resuspension and the corresponding nutrient release from the sediment–water interface, played an important role in intensifying cyanobacterial bloom expansion [9,10].

After blooms, the massive cyanobacterial biomass aggregates into scum and is then degraded by indigenous heterotrophs [11]. Large fractions of cyanobacterial organic matter are released from the biomass as dissolved organic matter (DOM) mainly via three major decline steps, which were proposed to describe the bloom decline process: scum disaggregation, colony settlement, and cell lysis in colonies [12]. Various bacteria of myxobacterium and the genera *Alcaligenes* and *Pseudomonas* and the *Cytophaga*/*Flavobacterium* group from aquatic environments were found to be lysed *Microcystis*, which is closely related to the degradation of biomass produced by cyanobacteria [13,14]. This algal-released DOM can be referred to as bulk algal-derived DOM. When large complex algal-derived DOM enters the aquatic system, it may cause oxygen depletion and cyanobacterial toxin secretion by biological activities, resulting in the appearance of a “black water mass,” which may pose serious risks to aquatic plants, fish, and oxygen-sensitive invertebrates and cause mass mortality events [15,16,17]. The high frequency formation of a massive bloom composed of cyanobacteria of the genus *Microcystis*, which can produce potent hepatotoxins, neurotoxins, and dermatoxins and promote tumor formation (i.e., microcystins), and they finally cause damage to the liver [18,19].

Several studies have suggested that bulk algal-derived DOM is closely related to the biogeochemical cycle and energy flow of lake ecosystems, which indirectly impact human health and the dynamics and driving factors of cyanobacterial blooms [20,21,22]. For example, Bittar et al. [23] confirmed that extracellular and intracellular DOM were produced in axenic cultures by *Microcystis aeruginosa*, which can effectively increase the biolabile DOM (BDOM) to bacterial growth and activity in lake waters for timescales of minutes to days. Moreover, Lee et al. [24] found that algal-derived BDOM can be converted into recalcitrant organic matter (such as humic substances), which remains in the water column for a long period. On the one hand, this freshly produced BDOM would strongly influence its binding properties to heavy metals and thus their fate, mobility, and toxicity in aquatic environments; especially, the algal-derived polymeric component increases the coagulation and sedimentation rates of colloidal material and associated metals, and thus brings hidden potential dangers to benthic healthy [25,26]. On the other hand, after long-term biodegradation, progressive accumulation of algal-derived recalcitrant DOM, as disinfection by-product precursor compounds, is transformed into trihalomethanes and haloacetic acids (i.e., carcinogenic and mutagenic disinfection byproducts), which inevitably leads to increased costs of drinking water supply systems [27,28,29]. In addition, during the decay of cyanobacteria blooms in lake ecosystems, cyanobacteria toxins are introduced into water and could directly threaten human health [30,31].

Dissolved organic matter is a fundamental regulator of aquatic ecosystems, and algal-derived DOM often dominates the DOM composition in water during algal bloom decay in Lake Taihu [32]. Several authors have studied the production of DOM with unique compositions and stoichiometries that fuel successive trophic transfers of fixed carbon through initial assimilation by heterotrophs. However, their rates of processing phytoplankton DOM are controlled to a large extent by the biological availability of cyanobacterial exudate and lysate; thus, two different cyanobacteria fractions will have implications for lake microorganisms to facilitate the efficient turnover of the highly heterogeneous cycle of available DOM [33,34]. Microbial-mediated metabolism reactions have been demonstrated to transform DOM from labile to a relatively recalcitrant state. This process is associated with nutrient recycling, greenhouse outgassing, and microbial secondary production, affecting microbial succession, as reflected in the taxonomic composition and functional diversity, as well as the growth rates of specific groups [35,36,37,38,39].

For lake ecosystems, especially eutrophic lakes, the nutrients needed to maintain the system largely depend on the internal circulation and regeneration of the system rather than external input [40]. Among them, the degradation of cyanobacterial organic matter and the release process of nutrients, which are important carriers of nutrients in water, may be important sources of active nutrients in water [41]. Algal-derived DOM has generally been used to represent the heterogeneous matrix for the entire decomposition of cyanobacterial blooms; however, most studies have focused primarily on the environmental behavior of bulk algal-derived DOM or the response of the interaction between bulk algal-derived DOM and water contaminants. However, the relationship between released DOM refers to the different fractions of cyanobacterial organic matter, and there have been few studies on reproduced nutrient and microbial secondary production (i.e., ecological effect) following the cyanobacterial cell lysis mechanism.

Therefore, the objective of this study was to gain new insight into the role of molecular weight-fractionated cyanobacterial organic matter on DOM mineralization, sequestration, and related nutrients. We set up a medium-volume, long-term incubation experiment and combined it with a fractionation procedure to investigate how DOM from lysed cyanobacteria cells impacts the variation of the microbial population, and how processing of algal-derived DOM impacts nutrient mineralization by comparative analysis. The results will provide data support for the ecological effects of cyanobacteria degradation.

## 2. Materials and Methods

### 2.1. Sample Collection and Pretreatment

Surface lake water and fresh algal scum of the phytoplankton *Microcystis* were collected at the aggregate stage of a cyanobacterial bloom along the shore of Lake Taihu (trestle area), China, in 2021 [42] (Appendix A). Samples (i.e., fresh algal scum) were stored at 4 °C and transported to the laboratory, where they were immediately filtered through a sterile 20-μm-pore nylon net (47-mm diameter, Merck Millipore, Ireland) to exclude the interference of the largest algal aggregates, protozoa, and non-living particulates in the microbial degradation system. This filtration for lake water through a 20-μm-pore-size filter (47-mm diameter, Merck Millipore, Cork, Ireland) was performed to separate particle-associated bacteria or large cells from small and free-living bacteria [43]. Then, the chlorophyll a (Chl a) concentration was measured (18.61 μg L^−1^) and considered the low biomass in summer. Chlorophyll a concentration was similar to the annual average in the lake center, and therefore, this was considered as a control set [44]. Meanwhile, harvested algal scum were concentrated through 20-μm bolting silk to partly dewater and remove obvious impurities; then, these samples with 90% moisture were defined as algal organic matter.

We considered the algal organic matter as a whole, which could be divided into two fractions by the freeze-thaw method. First, 10 mL algal organic matter-sterile water solutions with multiple batches (0.03 g mL^−1^ fresh weight) were shaken on a shaker, and we then obtained a destructed cyanobacterial complex mixture; the solids of these mixture were concentrated using a 0.2-μm membrane filter (47-mm diameter, PC, Merck Millipore, Cork, Ireland), and the filtrates were obtained. Destruction of cells in the above mixture was performed using the freeze-thaw method with three successive recycles, as described in a previous study [45]. The efficiency of cell destruction was confirmed using an optical microscope [46].

### 2.2. Batch Experiment

#### 2.2.1. Long-Term Microbial Degradation of Algal Organic Matter

At room temperature (20–25 °C) and under dark conditions, the cyanobacterial scum sampled from Lake Taihu shore and then pretreated as the 0.5 L algal organic matter-sterile water solution (0.03 g mL^−1^ fresh weight) was added to 9.5 L of filtered lake water containing bacterial communities and mixed in an acid-cleaned glass container to conduct a long-term (80-d) degradation experiment as the algal-derived DOM group. The purpose of setting this algal biomass concentration, i.e., ~1.5 g L^−1^ (fresh weight), was to simulate the real accumulation situation along the lakeshore as best as possible. The purpose of using dark incubation conditions was to prevent the possible growth of cyanobacteria that may have been present in filtered lake water. Therefore, the possible growth of cyanobacteria will interfere with the experimental results. Simultaneously, the long-term degradation of the filtered lake water without added cyanobacterial scum was set as the control group (i.e., natural lake water group). Both groups had three replicates, and all containers were covered tightly to avoid direct contact with the atmosphere. By collecting water samples at different intervals during the 80-d degradation process, the dynamic changes in planktonic bacterial abundance in lake water, and the concentrations and compositions of dissolved organic carbon (DOC) and nutrients were analyzed.

#### 2.2.2. Process Analysis of Microbial Degradation of Algal Organic Matter Using the Fractionation Procedure

The above-described long-term microbial degradation systems were repeated under the same controlled conditions in which 0.5 L of the extracted filtrates and concentrated solids, which were resuspended in an equal volume of sterile water, were added to the filtered lake water that served as the algal-derived filtrate and algal residual exudative organic matter groups. These two treatments were also conducted in triplicate in the dark for 80 d. As for the bulk algal-derived DOM and control groups, subsamples from the algal-derived filtrate and algal residual exudative organic matter groups were also collected on days 0, 1, 2, 4, 8, 12, 20, 31, 40, 52, 61, and 80 for analyses of planktonic bacterial abundance and determination of DOC and nutrient concentrations and compositions. Intriguingly, we observed that the water color of the algal-derived filtrate group changed dramatically (from brilliant blue to emerald green); therefore, an additional sample was collected on day 3.

### 2.3. Sample Analysis

#### 2.3.1. DOC and Chromophoric Dissolved Organic Matter (CDOM) Measurement

For the analysis of DOC concentration, 40 mL of water sample was filtered through pre-combusted (450 °C for 4 h) 0.22-μm-pore-size glass microfiber filters (47-mm diameter, Shanghai Xingya Purification Material Factory, Shanghai, China). Then, the filtrates were collected with glass pipettes, placed into pre-combusted brown glass vials, and stored at −20 °C until analysis. Reference DOC standards (obtained using potassium hydrogen phthalate) served as an additional control to calibrate the instrument. The blank was deducted using Milli-Q water analysis before every five samples, and all samples (including the blank) were acidified to pH = 2 by 10% HCl until analysis. The average blank concentrations associated with the DOC measurement were approximately 0.06 mg L^−1^, and the analytic precision of the triplicate injections was ±3%.

After passing through pre-combusted 0.22-μm Millipore membrane filters (47-mm diameter, Merck Millipore, Cork, Ireland), approximately 100 mL filtrate was used for CDOM absorbance and fluorescence measurements. CDOM absorbance was measured over the 200–800 nm range (1 nm increments) in a 5-cm quartz cell using a Shimadzu UV-2450 PC UV–vis recording spectrophotometer. A fluorescence spectrophotometer (Hitachi F-7000, Tokyo, Japan) with a scanning speed of 2400 nm min^−1^ was employed to measure the excitation-emission matrix (EEM). The main components of fluorescence dissolved organic matter (FDOM), i.e., the fluorescent fraction of CDOM, were analyzed using a fluorescence spectroscopy technique coupled with parallel factor (PARAFAC) analysis [47]. During the experiments, a total of 150 EEM spectra were obtained for PARAFAC analysis. MATLAB (MathWorks, Natick, MA, USA) and the DOM Fluor toolbox (http://www.models.life.ku.dk/ accessed on 6 May 2022) were employed for data analysis [48,49]. Further details on the PARAFAC-EEM analysis of FDOM have been described previously [50] and are displayed in Appendix A.

During analysis of the composition parameters of CDOM, the characteristic parameters of the CDOM absorption coefficient [a (355)] and spectral slope for the interval of 300–500 nm [S_300–500_] were employed to estimate the concentration and composition dynamic of CDOM, respectively [47]. Moreover, high S_300–500_ values denote a high extent of recalcitrant and a low degree of molecular weight [51].

#### 2.3.2. Nutrient Concentrations

Inorganic nitrogen (i.e., nitrate, nitrite, and ammonium) and inorganic phosphorus (i.e., phosphate) concentrations were measured using a continuous flow analyzer (San + +, SKALAR, Breda, The Netherlands). Total dissolved nitrogen (TDN) and total dissolved phosphorus (TDP) concentrations were analyzed using combined persulfate digestion followed by spectrophotometric measurements [52].

#### 2.3.3. Bacterial Abundance (BA)

Samples for analyses of BA were preserved with a final concentration of 0.5% glutaraldehyde and stored at −80 °C. BA was measured using an LSRFortessa flow cytometer (BD Biosciences, San Jose, CA, USA) by staining with SYBR Green I. Bacteria were enumerated according to a previously described method [53].

### 2.4. Statistical Analysis

The linear regression model was used by OriginPro 8.1 software (OriginLab, Northampton, MA, USA) to characterize the following relationships: (1) among the free-living bacterial abundance and CDOM absorption coefficient at 355 nm for the entire process in the bulk algal-derived DOM group; (2) among free-living bacterial abundance and the main components of FDOM identified by PARAFAC analysis for the entire process in bulk algal-derived DOM, algal-derived filtrate, and algal residual exudative organic matter groups; and (3) among free-living bacterial abundance and the fluorescence intensity of humic-like components for the day 40–80, day 20–80, and day 31–80 phases in the bulk algal-derived DOM, algal-derived filtrate, and algal residual exudative organic matter groups, respectively. To examine the significance of the temporal changes in the main components of FDOM identified by PARAFAC analysis during the experiment, we mainly aimed at comparing the bulk algal-derived DOM and control groups, and the algal-derived filtrate, and algal residual exudative organic matter groups. One-way analysis of variance (ANOVA) was performed using the data analysis function of OriginPro 8.1 software.

## 3. Results and Discussion

### 3.1. Algal Organic Matter Analysis

The initial properties of the selected algal-derived DOM and its different fractions are summarized in Table 1. In brief, the initial bulk algal-derived DOM was composed of 67.15% carbon, 26.38% nitrogen, and 6.47% phosphorus, whereas the initial algal-derived filtrate and residual exudative organic matter were composed of 77.86% and 58.33% carbon, 19.55% and 39.06% nitrogen, and 2.59% and 2.60% phosphorus, respectively. These values showed that the carbon component of the algal-derived filtrate was higher than that of the bulk algal-derived DOM; the nitrogen component of algal residual exudative organic matter was higher than that of the bulk algal-derived DOM. In contrast, the phosphorus components of the algal-derived filtrate and algal residual exudative organic matter were lower than that of the bulk algal-derived DOM. After freeze-thaw treatment, the CDOM absorption coefficients at 355 nm for initial algal-derived filtrate and algal residual exudative organic matter were approximately 0.79 and 0.63 times lower than that of initial bulk algal-derived DOM, respectively. Obviously, the FDOM composition of the initial bulk algal-derived DOM contained only two components (i.e., protein-like C1 component and humic-like C2 component), and C1 was the main fraction in the bulk algal-derived DOM. Surprisingly, through the comparison among the FDOM composition for initial bulk algal-derived DOM, filtrate, and algal residual exudative organic matter, it appeared that the humic-like C2 component was enriched in the algal-derived filtrate, which was approximately > 6 times higher than that of the other two groups.

### 3.2. Release and Microbial Degradation of Algal-Derived DOC

The dynamic changes in the DOC concentration of the bulk algal-derived DOM, algal-derived filtrate, and algal residual exudative organic matter during the release and microbial degradation processes are shown in Figure 1. In detail, DOC was released from the cyanobacterial scum, resulting in a linear increase in the DOC concentration from 8.36 ± 0.32 mg L^−1^ to a peak of 10.68 ± 0.17 mg L^−1^ during the first 4 d. Then, the DOC concentration decreased from 10.68 ± 0.17 to 6.86 ± 0.86 mg L^−1^ from day 4–40 with a mean reduction rate of 0.11 ± 0.02 mg L^−1^ d^−1^. Subsequently, the DOC concentration decreased to 4.60 ± 0.09 mg L^−^^1^ at the end of the experiment with a mean reduction rate of 0.06 ± 0.00 mg L^−1^ d^−1^ (Figure 1a). Therefore, according to the dynamic changes in the DOC concentration, the 80-d degradation process was clearly divided into three stages: the DOC rising (DR) stage (4 days); rapid DOC decline (r-DD) stage (36 days); and slow DOC decline (s-DD) stage (40 days). In contrast, no significant fluctuations in the DOC concentration were observed in the lake water; rather, it had a relatively narrow range from 3.34 ± 0.03 to 4.48 ± 0.21 mg L^−1^ and a mean of 3.98 ± 0.42 mg L^−1^. Meanwhile, we determined that the DOC mineralization rate was 56.88 ± 1.26% from the time point of maximum DOC release to the end of the experiment. As shown in Figure 1b, the initial DOC concentrations were 10.81 ± 0.47 mg L^−1^ and 8.68 ± 0.33 mg L^−1^ in the algal-derived filtrate and the algal residual exudative organic matter treatments, respectively. Specifically, the algal-derived filtrate treatment caused a rapid and markedly large increase in the DOC concentration; however, the DOC concentration was slightly lower in the algal residual exudative organic matter group at the initial time point. Throughout the microbial degradation process, the DOC concentration showed a rapid decrease in the first 20 d (from 10.81 ± 0.47 to 5.57 ± 0.33 mg L^−^^1^), followed by a near-continual slow decrease with an overall concentration of 4.44 ± 0.52 mg L^−1^ for the algal-derived filtrate group; in short, the DOC mineralization rate of the entire process was 64.56 ± 0.32%. In another algal residual exudative organic matter group, the DOC concentration showed a linear decrease over the entire experimental process by 51.83 ± 5.12% (i.e., the DOC mineralization rate). Finally, the DOC concentrations were 3.83 ± 0.18 and 4.17 ± 0.30 mg L^−1^ in the algal-derived filtrate and the algal residual exudative organic matter treatments, respectively, on day 80.

### 3.3. CDOM Absorption and Spectral Slope in Different Algal-Derived DOM

Generally, CDOM is largely responsible for the optical properties of most natural waters, and as a tracer of algal-derived DOM is valuable for elucidating the dynamic changes of DOM. The specific changes in the CDOM absorption coefficient [a (355)] of the bulk algal-derived DOM, algal-derived filtrate, and algal residual exudative organic matter during the 80 d processes are shown in Figure 2. In detail, in the bulk algal-derived DOM group, the a (355) increased from 24.35 ± 1.71 to 29.24 ± 0.37 m^−1^ within the first day, and then decreased to 15.93 ± 0.82 m^−1^ on day 4, with a reduction rate of 4.46 ± 0.60 m^−1^ d^−1^ from day 1–4. During the process from day 8–80, the a (355) fluctuant decreased from 8.11 ± 0.98 m^−1^ to 3.58 ± 0.11 m^−1^. After repeated measurement, we found that an abnormal increase to 8.72 ± 0.23 m^−1^ occurred suddenly on day 40. Compared to the lake water, the a (355) exhibited relatively small variations and remained at the mean level (i.e., 2.40 ± 0.45 m^−1^) during the entire experimental process. As shown in Figure 2b, after the different amended cyanobacterial organic matter fractions were added, we found that it led to large increases in a (355) in algal-derived filtrate and algal residual exudative organic matter groups; i.e., a (355) increased by 7.19 ± 0.04 and 5.75 ± 0.15 times compared with the initial lake water value, respectively. Then, a (355) degradation showed similar exponential decay patterns in both the algal-derived filtrate and algal residual exudative organic matter groups, and their intensities remained steady at approximately 5.20 ± 0.47 and 4.22 ± 0.81 m^−1^, respectively, with time after decreasing rapidly in the first 20 and 31 d, respectively.

The CDOM spectral slope for the 300–500 nm [S_300–500_] interval was employed to estimate the composition dynamic of CDOM. Moreover, high S_300–500_ values denote a high extent of recalcitrant and a low degree of molecular weight [51]. Throughout the entire bulk algal-derived DOM degradation process, the spectral slope S_300–500_ gradually increased from 6.69 ± 0.06 μm^−1^ to 15.71 ± 0.49 μm^−1^; it indicated the transformation from high-molecular-weight labile CDOM into low-molecular-weight recalcitrant CDOM. In the control group, Figure 3a shows that the spectral slope S_300–500_ had a relatively narrow range variation from 11.04 ± 0.30 to 13.69 ± 0.07 μm^−1^ within the first 40 d, and that the index then increased to 17.24 ± 0.20 at the final time point. Notably, at the s-DD stage, the S_300–500_ index was near the mean level of the control group; this implied freshly cyanobacterial scum-released CDOM underwent long-term biotransformation to perform some molecular weight CDOM similar to cyanobacterial scum-free lake water. For the algal-derived filtrate group (Figure 3b), the spectral slope S_300–500_ increased by 5.12 ± 0.04 μm^−1^, i.e., from 8.10 ± 0.06 to 13.22 ± 0.06 μm^−1^, over the course of the experiment. Another algal residual exudative organic matter group exhibited highly dynamic characteristics because the spectral slope S_300–500_ increased by 9.20 ± 0.34 μm^−1^ from 5.44 ± 0.04 to 14.63 ± 0.35 μm^−1^ over the course of the experiment (Figure 3b). Therefore, the initial molecular weight of the algal residual exudative organic matter group was higher than that of the algal-derived filtrate and bulk algal-derived DOM groups; however, after the 80-d experiment, the molecular weight of DOM was converted to the same level for the three groups.

### 3.4. EEMs in Different Algal-Derived DOM

The characterization of DOM by 3D fluorescence spectroscopy was considered a reliable parameter for observing the entire degradative process for bulk algal-derived DOM and natural lake water groups (as a control). As shown in Figure 4a–c, we found that the EEMs of CDOM released from cyanobacterial scum at two major protein-like fluorescence peaks decreased with time and reached a low level at the end of the experiment, which was characterized by being highly dynamic. Comparatively, for the natural lake water group, a conservative distribution of inherent CDOM peaks at the two similar protein-like fluorescence peaks was observed; however, its fluorescence intensities and the range of increasing–decreasing fluctuations were both lower than those of the bulk algal-derived DOM group. This result was consistent with those reported previously [54], whereby cyanobacteria survival and mortality played important roles in shaping the optical properties of many natural waters. Moreover, decayed cyanobacteria led to protein-like CDOM production, which was an important source of biodegradable DOC and contributes to the biogeochemical cycles of aquatic ecosystems [55]. Comparing the EEMs of the algal-derived filtrate and the algal residual exudative organic matter groups, as shown in Figure 5, we found that high-intensity protein peaks dominated the entire spectra, and that after 80-d degradation, their fluorescence intensities were greatly reduced. Finally, the variability in the fluorescence properties of CDOM of the two groups was similar to that of the control group, as determined by EEMs. More specifically, the intensities of humic peaks were almost consistent with the intensities of protein peaks, which only appeared in the algal-derived filtrate group at the initial time point. This suggested humic-like components direct from cyanobacteria cell lysis, and that this component has low molecular weight characteristics, which is indicated by the higher values of S_300-500_ than the algal colloidal exudative organic matter group during the initial 20 days.

### 3.5. FDOM Components in Different Algal-Derived DOM

FDOM was employed as a proxy for DOM to study the dynamic changes in DOM composition [56]. The FDOM components were characterized with three-dimensional EEM spectroscopy coupled with the PARAFAC analysis technique, and total EEM collection of the algal-derived dissolved organic matter; its different molecular fractions and lake water were modeled with PARAFAC using MATLAB with the DOMFluor toolbox. Furthermore, four distinct FDOM components (i.e., C1–C4) were identified; in detail, the ex|em wavelengths were 225 (275–280)|332 nm, 265 (360)|452 nm, 230 (275)|320 nm, and 235 (305)|348 nm, respectively. Based on their ex|em wavelengths, C1, C3, and C4 represented protein-like fractions and C2 represented a humic-like fluorophore. During the bulk algal-derived DOM degradation process, the dynamic changes in the fluorescence intensity of the protein-like component C1 exhibited three stages, which corresponded to the stage characteristics of dynamic changes in the DOC concentration (Figure 1). Furthermore, its fluorescence intensity decreased periodically at 0.89 ± 0.11 R.U. (mean values) in the DR stage, at 0.21 ± 0.07 R.U. (mean values) in the r-DD stage, and the final fluorescence intensities fell to zero (mean values) in the final s-DD stage (Figure 6a). Interestingly, the change in the humic-like component C2 showed an opposite trend compared with that of the protein-like component C1 (Figure 6b). The C2 component increased gradually throughout the degradation process, except for a sudden increase on the first day (Figure 6b). However, the two fluorescence components (i.e., C1 and C2) in the control group exhibited no significant changes throughout the entire degradation process (Figure 6a,b). Moreover, we found that the variation patterns of the C1 and C2 components, when comparing the bulk algal-derived DOM and natural lake water groups, exhibited significant changes (one-way ANOVA, *p* < 0.05); therefore, these two components also should focus on fluorescence variety with time for the algal-derived filtrate and the algal residual exudative organic matter groups. Regarding the two other protein-like C3 and C4 components, their changes tended to be consistent, and both remained at relatively low levels compared to the biodegradable protein-like C1 component (Figure 6c,d).

Similarly, four FDOM components (C1, C2, C3, and C4) were identified in the algal-derived filtrate and the algal residual exudative organic matter groups. Based on our previous results regarding dynamic changes in the FDOM compositions of bulk algal-derived DOM and control groups, the intensity changes of the protein-like component C1 and humic-like component C2 were comprehensively examined again in algal-derived filtrate and algal residual exudative organic matter groups. Notably, the dynamic variations in fluorescence intensity of C1 and C2 underwent clear changes, which were also reflected by the three-stage characteristics over the entire degradation period (Figure 7a,b). In the algal-derived filtrate group, the fluorescence intensity of C1 decreased from 0.58 ± 0.03 to 0.47 ± 0.01 R.U. with a decrease rate of 0.04 R.U. d^−1^ from day 0–3. Subsequently, its fluorescence intensity decreased to 0.05 ± 0.02 R.U. at day 20, with a decrease rate of 0.02 R.U. d^−1^, and stabilized at 0.02 ± 0.02 R.U. from day 20–80. In contrast, the fluorescence intensity of C2 increased from 0.07 ± 0.01 to 0.10 ± 0.0004 R.U. for the period of day 20–80 (Figure 7a,b). Similarly, in the algal residual exudative organic matter group, the protein-like component C1 decreased rapidly from 1.02 ± 0.01 to 0.51 ± 0.04 R.U. with a decrease rate of 0.13 R.U. d^−1^ for the period of day 0–4, then decreased slowly to 0.04 ± 0.00 R.U. at day 31, with a decrease rate of 0.01 R.U. d^−1^, and reached near-constant intensities with a mean level of 0.02 ± 0.01 R.U. in the day 31–80 phase. The fluorescence intensity of C2 increased from 0.04 ± 0.003 to 0.08 ± 0.0003 R.U. in the day 31–80 phase (Figure 7a,b). Curiously, in the algal-derived filtrate, we found that the humic-like component C2, which is always used as an indicator of recalcitrant DOM [57], was found at much higher fluorescence signal levels when compared to the algal residual exudative organic matter and bulk algal-derived DOM samples on day 0. Furthermore, its fluorescence intensity remained high (~0.25 ± 0.04 R.U.) for the period of day 0–3, and suddenly decreased to a lower level of 0.13 ± 0.05 R.U. for the period of day 3–20 (Figure 7b). Zuo et al. [58] identified 6-L-biopterins and their glucosides as candidate structures for consistently occurring algae-derived humic-like fluorophores (Em 440–460 nm) during the cyanobacterial strain *Microcystis aeruginosa* degradation experiment under simulated natural conditions; the environmental concentrations of 6-L-biopterin (without counting any other derivatives) ranged from 0.20–2.78 μg L^−1^ in five lakes in China. Additionally, biopterin and its derivatives, which contributed to 55.5 ± 1.7% of fluorescence at Ex350/Em450 nm in FDOM, were found in a Lake Tai surface water sample [58]. In this study, C2 displayed two excitation maxima at 265 and 360 nm and one emission maxima at 452 nm, which was categorized as a humic-like peak; therefore, it may have indicated that the algae-derived characteristic humic-like fluorophores were associated with biopterin. Humic-like fluorophores are ubiquitous in algal-dominated freshwater and marine environments [59,60]. Through this research, we found that humic-like components rich in cyanobacterial filtrate could not accumulate in lake and coastal eutrophic water, which is inconsistent with research results from pelagic oligotrophic sea and deep-sea zones. For instance, Xie et al. [55] investigated the bioavailability of *Synechococcus*-derived organic matter by estuarine and coastal microbes during 180-d dark incubations, and found that humic-like C4 (ex|em wavelength was 250 (385)|484 nm) displayed recalcitrant DOM characteristics, and that its fluorescence intensity gradually increased over the entire incubation period. One possible explanation for this result is that the high decrease rate of humic-like fluorophores was supported by the high content of biodegradable protein-like components or nutrients as well as the metabolically active microbial populations in the eutrophic lake water [47]. In contrast, in oligotrophic environments, freshly released algal filtrates can hardly trigger the organic matter “priming effect” mechanism that stimulates the microbial degradation of humic-like components, whereas humic-like organic matter is often considered a potential tracer of recalcitrant DOC [61,62,63]. Lake eutrophication can result in algal blooms and water quality degradation, which affects the services provided by the lake ecosystem [64]. As discussed above, initial DOM compositions derived from algal-derived filtrates and algal-residual exudative organic matter were highly heterogeneous in terms of FDOM composition, whereby their FDOM compositions were consistent with each other after a long degradation period.

### 3.6. Variability of Nutrient Compositions in Different Algal-Derived DOM

Along with the release and microbial degradation of DOC derived from cyanobacterial scum, variations in inorganic nutrients (including NH_4_^+^, NO_2_^−^, NO_3_^−^, and PO_4_^3−^) and organic nutrients (including DON and DOP) were presented during the 80-d incubation (Figure 8c,d). In detail, in the bulk algal-derived DOM group, in terms of the variations of TDN and TDP, the TDN concentration gradually increased from 6.01 ± 0.36 to 12.38 ± 1.06 mg L^−^^1^ in the day 0–31 phase, then remained at 11.91 ± 0.96 mg L^−1^ during day 31–80 (Figure 8a); the TDP concentration gradually increased from 0.72 ± 0.03 to 1.10 ± 0.16 mg L^−^^1^ in the day 0–31 phase, then remained at 1.19 ± 0.10 mg L^−1^ for the day 31–80 phase. Three inorganic nitrogen compositions were measured, and the combined concentrations as dissolved inorganic nitrogen (DIN) and its variation are shown in Figure 8b, in which a linear increase can be seen from 3.73 ± 0.39 to 14.26 ± 1.69 mg L^−1^ with an increase rate of 0.29 ± 0.05 mg L^−1^ d^−1^ during the day 0–31 phase. Within the next 49 d, the DIN concentration increased to 13.29 ± 0.90 mg L^−1^ with a minimal increase rate of 0.01 ± 0.06 mg L^−1^ d^−1^. The variation of DIP followed the same pattern, whereby it increased from 0.28 ± 0.06 to 1.11 ± 0.09 mg L^−1^ with an increase rate of 0.03 ± 0.00 mg L^−1^ d^−1^ during the day 0–31 phase and was subsequently relatively stable within the range of 0.41–0.52 mg L^−1^. Meanwhile, in addition to leaching dissolved total nutrients from cyanobacterial biomass, there are many other nutrient metabolism processes, i.e., organic nitrogen and phosphorus degradation, and inorganic nitrogen and phosphorus production (Figure 8b,c). Moreover, ammonia was the most dominant form of DIN, with the conversion of cyanobacteria-derived organic nitrogen to inorganic nitrogen (Figure 8b,d). In the day 40–80 phase, the DON and DOP concentrations tended to stabilize and maintain mean concentrations of 0.34 ± 0.46 and 0.07 ± 0.08 mg L^−1^, respectively; specifically, the dominant form of DIN was converted into nitrate (Figure 8b–d). In the control group, the range of changes and contents in the dissolved nutrient composition were relatively gently and small, and the most dominant DIN (i.e., ammonia) during the initial 4 d was also replaced by nitrate over the course of the entire process (Figure 8e–h).

In the algal-derived filtrate group, the initial TDN and TDP concentrations were 6.46 ± 0.24 and 0.62 ± 0.01 mg L^−1^, respectively, during the first 3 d and then decreased to 5.51 ± 0.18 and 0.41 ± 0.01 mg L^−^^1^, respectively. For the following 17 d, TDN and TDP concentrations exhibited linear increases to 6.93 ± 0.02 and 0.63 ± 0.05 mg L^−1^, respectively; finally, the concentrations remained constant at 7.04 ± 0.41 and 0.67 ± 0.02 mg L^−1^, respectively (Figure 9a). In contrast, in the algal-derived filtrate group, the TDN concentration displayed a similar variation pattern to that of the TDP concentration. For the initial 12 d, TDN and TDP concentrations were stable at 4.85 ± 0.40 and 0.27 ± 0.03 mg L^−^^1^, respectively. After 20 d, TDN and TDP concentrations dramatically increased to 6.66 ± 0.51 and 0.44 ± 0.05 mg L^−1^, respectively (Figure 9e). Two processes of total dissolved nutrient release and dissolved organic nutrient conversion into dissolved inorganic nutrients were detected from 0–20 d in both groups. Ammonia was the only form of DIN that was present. Moreover, in the following 60 d, the variations in the DIN, DON, DIP, and DOP concentrations exhibited roughly stable levels; notably, the majority of DIN during this period was replaced by nitrate, and a high nitrate concentration was maintained (Figure 9b–d,f–h).

### 3.7. Variability of Free-Living Bacterial Abundance in Different Algal-Derived DOM

For the degradation of the bulk algal-derived DOM process, the change of free-living bacterial abundance showed highly dynamic characteristics and could be roughly divided into three stages, which correspond with the periodic transformation of different biological availability of algal-derived DOM (labeled Stage DR, r-DD, and s-DD). In the bulk algal-derived DOM group, the free-living bacterial abundance increased by 0.48 ± 0.71 × 10^7^ cells mL^−1^ (from 4.22 ± 0.28 × 10^7^ to 4.70 ± 0.51 × 10^7^ cells mL^−1^) in phase DR. Then, the free-living bacterial abundance decreased to 1.23 ± 0.31 × 10^7^ cells mL^−1^ at the end of phase r-DD and remained with relatively small variations in the range from 0.41 × 10^7^ to 1.77 × 10^7^ cells mL^−1^ in the following days (i.e., phase s-DD). In contrast, in the control lake water group, no significant fluctuations in bacterial abundance were observed; the range of change was between 0.06 × 10^7^ and 0.98 × 10^7^ cells mL^−1^ (Figure 10a).

The DOC concentration increased sharply and reached its maximum in only 4 d during the DR stage. Concurrently, the bacterial abundance also increased. This indicated that the release of DOC from decaying cyanobacteria was relatively rapid, and that the freshly released DOC was strongly favored by bacteria. During this stage, bacteria abundance slightly increased, indicating that a portion of the released DOC was utilized and transformed into bacterial biomass, or possibly some originally attached bacteria dispersal from dead algal biomass into the water. Subsequently, the DOC concentration began to decline, entering the r-DD stage. The decrease in DOC concentration may have been due to the slowdown in DOC release from the cyanobacterial scum not maintaining the same rate as DOC consumption by the abundant bacteria, or the life strategy of these bacteria undergoing changes [34]. The relatively rapid decline process in DOC continued for 5 weeks. This showed that a portion of the DOC released by the decaying algal scum was labile DOC and was easily utilized by bacteria. Indeed, in the eutrophic environment of Lake Taihu, lake water quality usually deteriorates within approximately 1 month after decomposition of *Microcystis* blooms [65]. One important reason may have been that, while utilizing the rich labile DOC released from decaying algal scum, the heterotrophic bacteria consumed oxygen and released CO_2_, resulting in decreased dissolved oxygen and impacts on the nutrient status of the surrounding water [66,67]. After the r-DD stage, during the remaining 40 d, the remaining DOC had lower bioavailability to bacteria, and partly the DOC fraction may have been in recalcitrant states [68]. Notably, bacterial abundance reached its maximum on day 2, decreased rapidly in the next 38 d, and then decreased at a slower rate in the final 40 d. This may suggest that partly DOC components become humified, which barely promotes the growth of free-living bacteria, leading to the death of some bacteria. Compared to the control group, the positive linear correlation between free-living bacteria abundance and DOC concentration for the entire process (*R*^2^ = 0.63, *p* < 0.01) further suggested that algae-released DOC stimulated microbial growth and activities [29]. The slope of significantly positive relationships between bacterial abundance and DOC concentration were found in the r-DD stage to be almost three times higher than the s-DD stage; this preliminarily reflected that the availability of DOM components and the bacterial responses to phytoplankton-derived carbon had changed. Furthermore, we found a relatively strong linear correlation between free-living bacteria abundance and CDOM absorption coefficient at 355 nm for the entire process (*R*^2^ = 0.68, *p* < 0.001), indicating that the microbial biodegradation process had transformed the autochthonous DOM from chromophoric components to nonchromophoric species. Furthermore, it showed that the growth and mortality of some free-living bacteria responded to the transformation of DOM, and the increasing changes in the spectral slope for the interval of 300–500 nm (i.e., S_300–500_), indicating the production of a large amount of high-molecular-weight CDOM through microbial conversion to lower molecular weight CDOM. By focusing on changes in the free-living bacteria abundance and the fluorescence intensities of four FDOM components, only the protein-like C1 fraction showed a significant and positive relationship with free-living bacteria abundance (*R*^2^ = 0.85, linear regression, *p* < 0.001), indicating that the protein-like C1 may be a critical bioavailable component and energy source for microbes in the lake. At the initial time point of the s-DD period, the CDOM absorption coefficient at 355 nm exhibited an abnormal increase on day 40. Concurrently, we found that the humic-like C2 component increased linearly and was weakly negatively correlated with the change in bacterial abundance (*R*^2^ = 0.24, *p* > 0.05), in contrast to the change in the protein-like C1 component, during the last s-DD stage. Therefore, we infer that the remnants of dead bacteria (i.e., necromass) and bacterial secretions may have made an important contribution to the accumulation of the humic-like C2 component. Therefore, we can conclude that the lysis of cyanobacterial scum by microbes liberates DOM into lake water, and the DOC concentration and composition in water are closely related to the abundance of planktonic bacteria.

For the algal-derived filtrate and algal-residual exudative organic matter groups, the dynamic change patterns in free-living bacterial abundance also showed three different characteristics among stages of the long-term degradation process (Figure 10b). At the beginning, bacterial abundances of these two treatments were 1.62 ± 0.17 × 10^7^ and 2.45 ± 0.29 × 10^7^ cells mL^−1^, respectively. These initial values were at much higher levels than the lake water blank sample that had been filtered through 20-μm pore size membranes; however, the sum of the two values was slightly less than that of the bulk algal-derived DOM group, with approximately 4% loss due to experimental operations. In the algal-derived filtrate group, the free-living bacterial abundance increased considerably in response to the addition of the carbon sources and reached 5.08 ± 0.15 × 10^7^ cells mL^−1^ on day 3. A sharp decline in the bacterial abundance was observed from day 3 to day 20 (i.e., 0.54 ± 0.05 × 10^7^ cells mL^−1^), and then there was a gradual decrease to 0.25 ± 0.01 × 10^7^ cells mL^−1^ on day 80. In the algal-residual exudative organic matter group, bacterial abundance showed an increase over the first 4 d to 5.23 ± 0.09 × 10^7^ cells mL^−1^, followed by a rapid decrease during the following 28 d (i.e., 1.09 ± 0.17 × 10^7^ cells mL^−1^) and near-constant abundances after day 31 to the end of the experiment with a mean abundance of 0.69 ± 0.33 × 10^7^ cells mL^−1^.

Significantly positive linear relationships between free-living bacterial abundance and protein-like C1 fluorescence were also found in the algal-derived filtrate group (*R*^2^ = 0.78, *p* < 0.001) and algal residual exudative organic matter group (*R*^2^ = 0.59, *p* < 0.01) throughout the entire experimental period. This indicated that labile algal-derived filtrate or algal residual exudative organic matter derived from algal degradation became an important factor in controlling bacterial abundance, which would play an important role in microbial food webs and carbon cycling during the decay of algal blooms through the bacterial enzymolysis mechanism [69]. At the final stable phase of the two groups (algal-derived filtrate group, day 20–80; algal residual exudative organic matter group, day 31–80), the humic-like C2 component accumulated with the decrease in free-living bacterial abundance, and free-living bacterial abundance showed a significant negative linear relationship with humic-like C2 fluorescence (algal-derived filtrate group, *R*^2^ = 0.63, *p* < 0.05; algal residual exudative organic matter group, *R*^2^ = 0.96, *p* < 0.01).

## 4. Conclusions

In this work, the labile fraction of fresh cyanobacterial organic matter could be rapidly utilized by microbes, and the subsequent microbial-mediated processes of cyanobacterial organic matter also drove the elemental cycling of carbon, nitrogen, and phosphorus. Specifically, we showed the gradual transformation of algal-derived DOC from a high-molecular-weight labile to a low-molecular-weight recalcitrant through the microbial process. Different dominant DOM compositions with bioavailable characteristics were present in different phases during bulk algal-derived DOM degradation, which were linked to the variation in the abundance of free-living bacteria. Furthermore, this process was coupled with the regeneration of different forms of inorganic nutrients. Algal-derived filtrate and algae residual exudative organic matter both made key contributions to the decomposition of algal blooms in eutrophic Lake Taihu. Additionally, the findings show that the carryover effects of *Microcystis* cyanobacteria blooms can exist for a long time, whereby approximately 7.45% and 18.60% of the released DOC could be converted into a stable state in the algal-derived filtrate and algal residual exudative organic matter groups, respectively.

## Figures and Tables

**Figure 1 ijerph-19-06981-f001:**
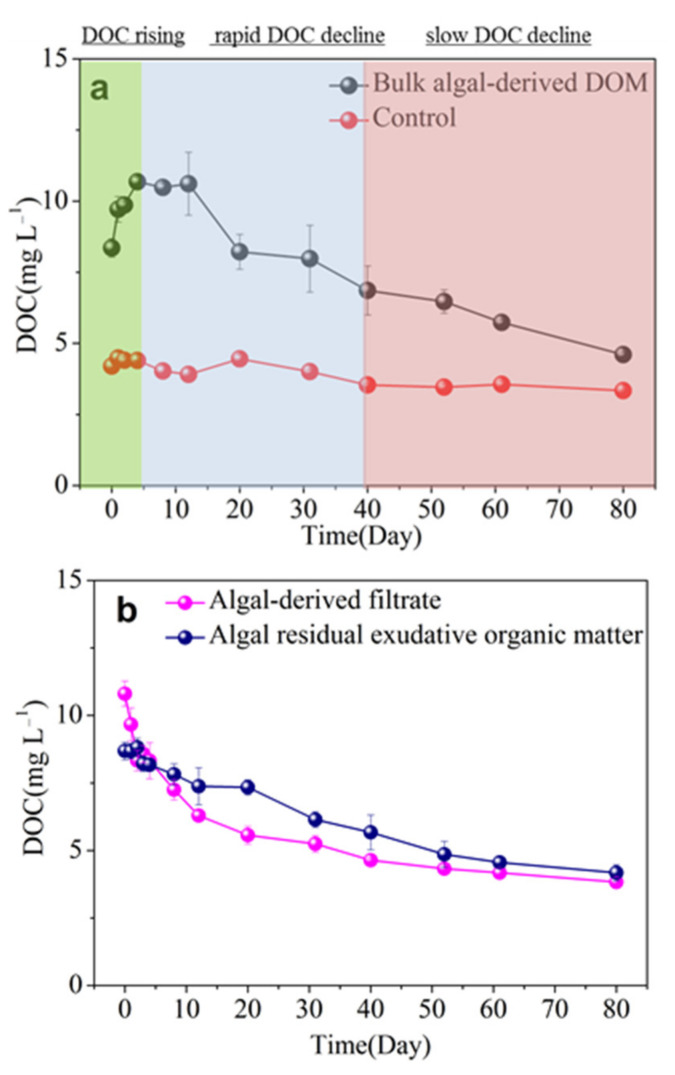
Dynamic changes in dissolved organic carbon (DOC) concentration during the 80-d degradation processes in (**a**) bulk algal-derived dissolved organic matter (DOM) group and natural lake water group (as control), and (**b**) algal-derived filtrate and algal residual exudative organic matter groups.

**Figure 2 ijerph-19-06981-f002:**
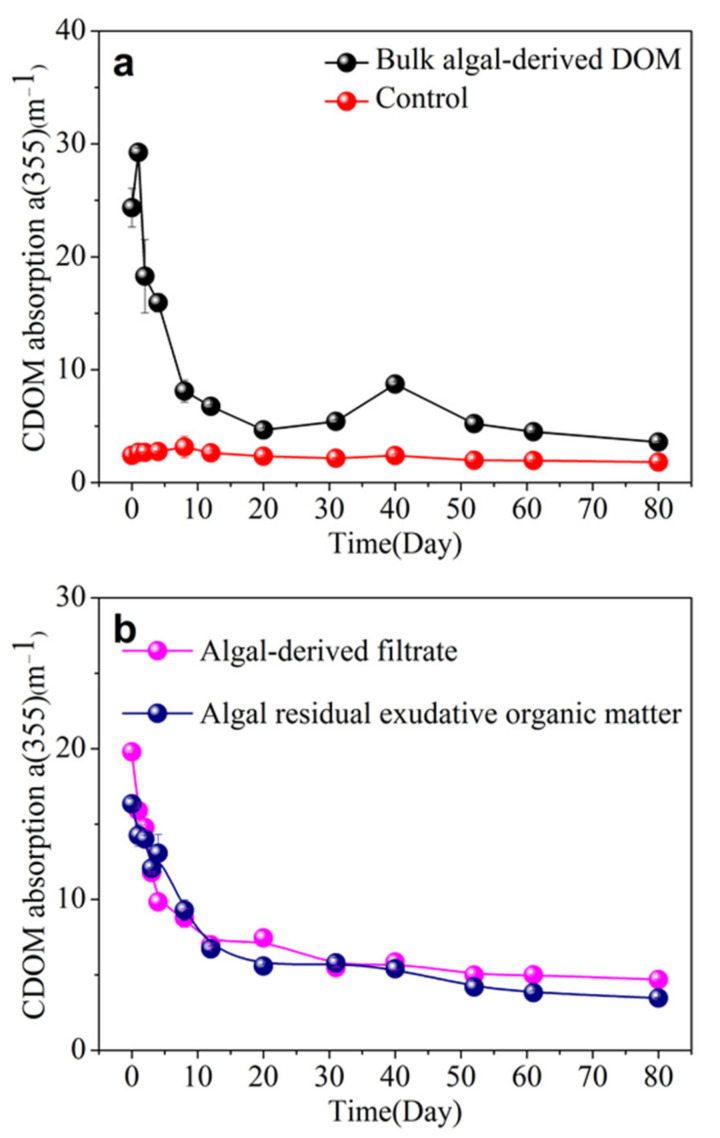
Dynamic changes in the chromophoric dissolved organic matter (CDOM) absorption coefficient at 355 nm [a CDOM (355)] during the 80-d degradation processes of (**a**) bulk algal-derived dissolved organic matter (DOM) and natural lake water groups (as control) and (**b**) algal-derived filtrate and algal residual exudative organic matter groups.

**Figure 3 ijerph-19-06981-f003:**
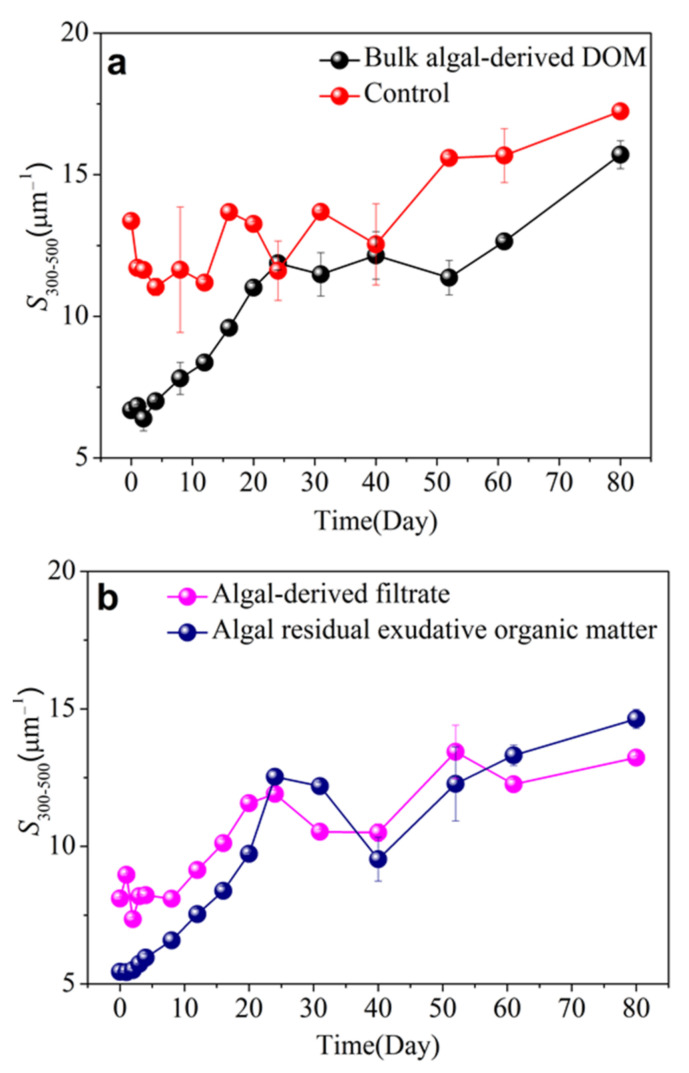
Dynamic changes in the chromophoric dissolved organic matter (CDOM) spectral slope for the 300–500 nm [S_300–500_] interval during the 80-d degradation processes of the (**a**) bulk algal-derived DOM and natural lake water groups (as control) and (**b**) algal-derived filtrate and algal residual exudative organic matter groups.

**Figure 4 ijerph-19-06981-f004:**
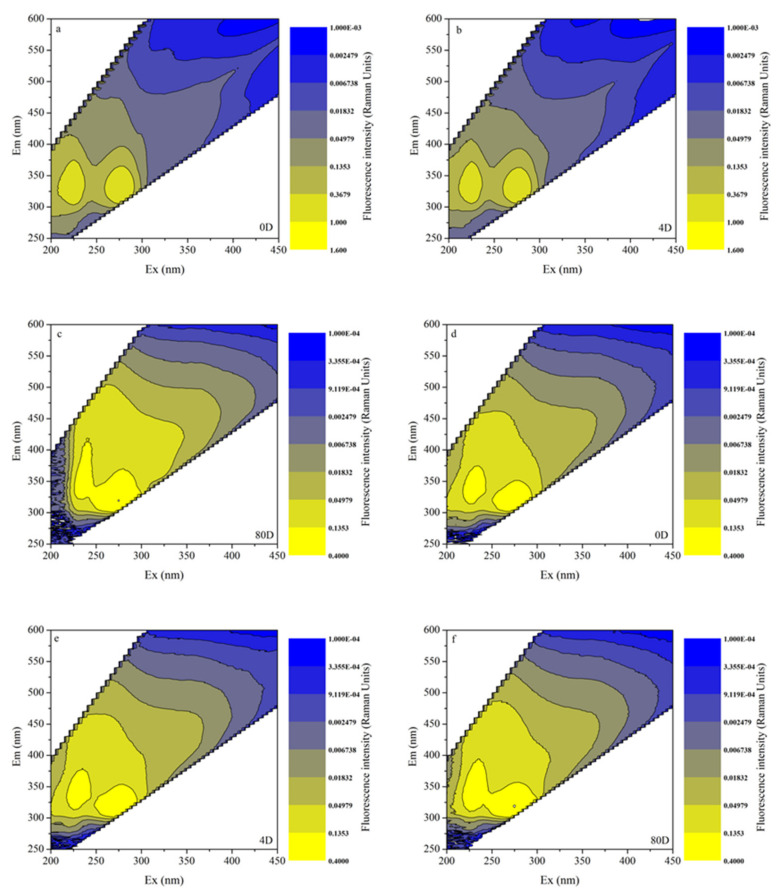
Chromophoric dissolved organic matter (CDOM) fluorescence properties variability by excitation-emission matrix spectra (EEMs) during the degradation processes of the bulk algal-derived dissolved organic matter (DOM) group on (**a**) day 0; (**b**) day 4; and (**c**) day 80, and in the natural lake water group (as control) on (**d**) day 0; (**e**) day 4; and (**f**) day 80.

**Figure 5 ijerph-19-06981-f005:**
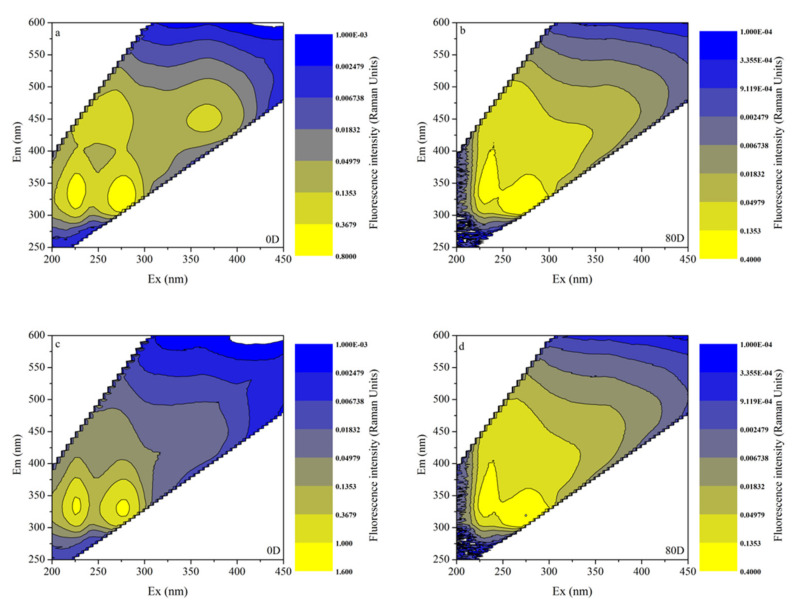
Chromophoric dissolved organic matter (CDOM) fluorescence properties presented by excitation-emission matrix spectra (EEMs) for the algal-derived filtrate group on (**a**) day 0 and (**b**) day 80, and for the algal residual exudative organic matter group on (**c**) day 0 and (**d**) day 80.

**Figure 6 ijerph-19-06981-f006:**
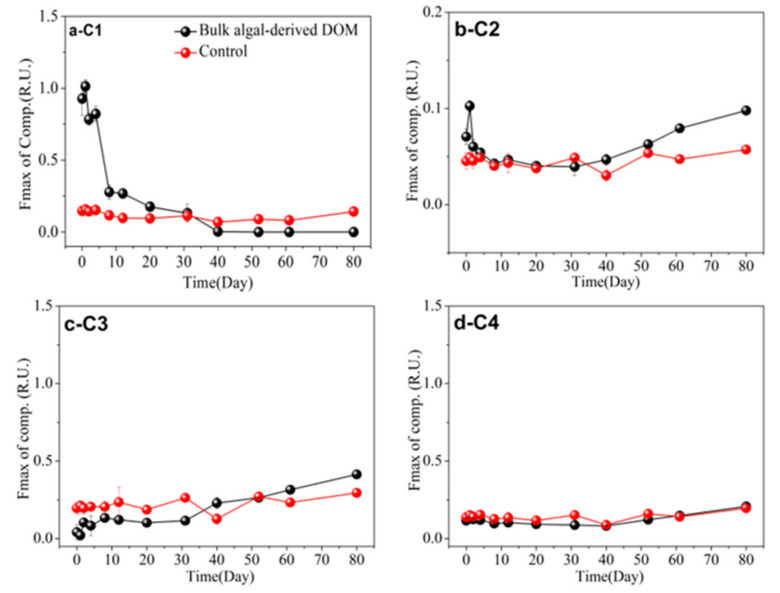
Dynamic changes in the main components of fluorescent dissolved organic matter (FDOM) during the 80-d degradation processes of the algal-derived DOM group and in the natural lake water group (as control). (**a**) C1, (**b**) C2, (**c**) C3, and (**d**) C4.

**Figure 7 ijerph-19-06981-f007:**
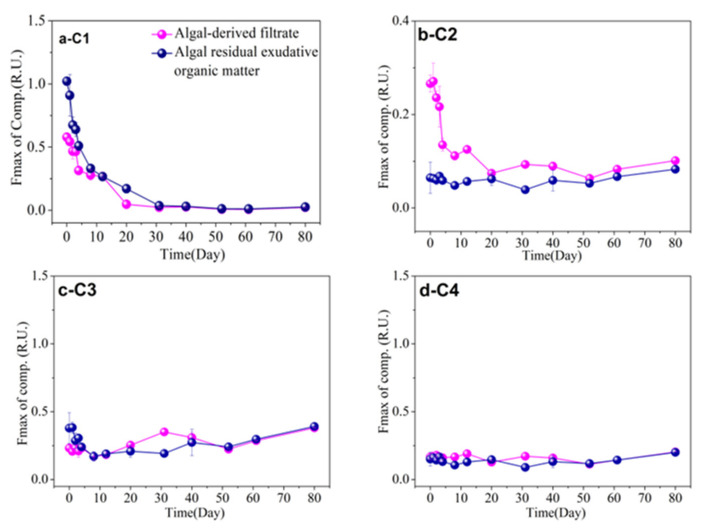
Dynamic changes in the main components of fluorescent dissolved organic matter (FDOM) during the 80-d degradation processes of algal-derived filtrate and algal residual exudative organic matter groups. (**a**) C1, (**b**) C2, (**c**) C3, and (**d**) C4.

**Figure 8 ijerph-19-06981-f008:**
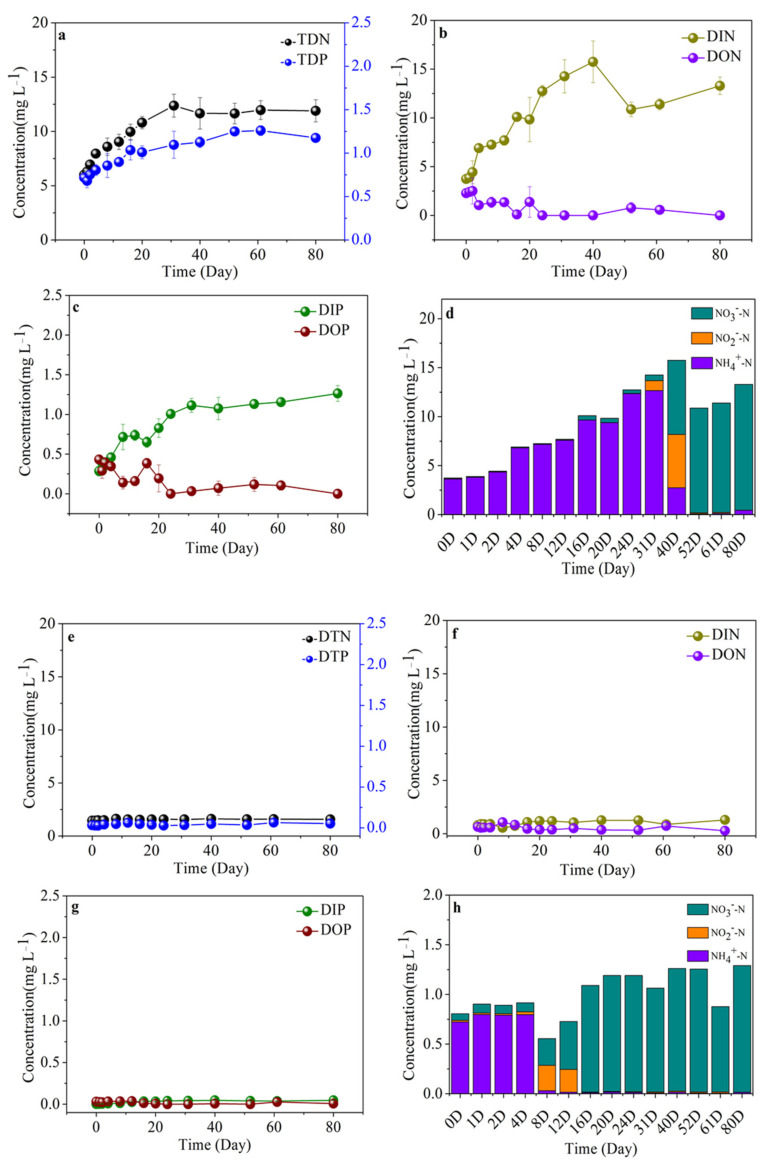
Variations in dissolved nitrogen and phosphorus concentrations and compositions during the 80-d degradation processes of the (**a**–**d**) bulk algal-derived dissolved organic matter (DOM) group and (**e**–**h**) natural lake water group (as control). TDN—total dissolved nitrogen, TDP—total dissolved phosphorus, DIN—dissolved inorganic nitrogen, DON—dissolved organic nitrogen, DIP—dissolved inorganic phosphorus, DOP—dissolved organic phosphorus, DTP—dissolved total phosphorus, DTN—dissolved total nitrogen.

**Figure 9 ijerph-19-06981-f009:**
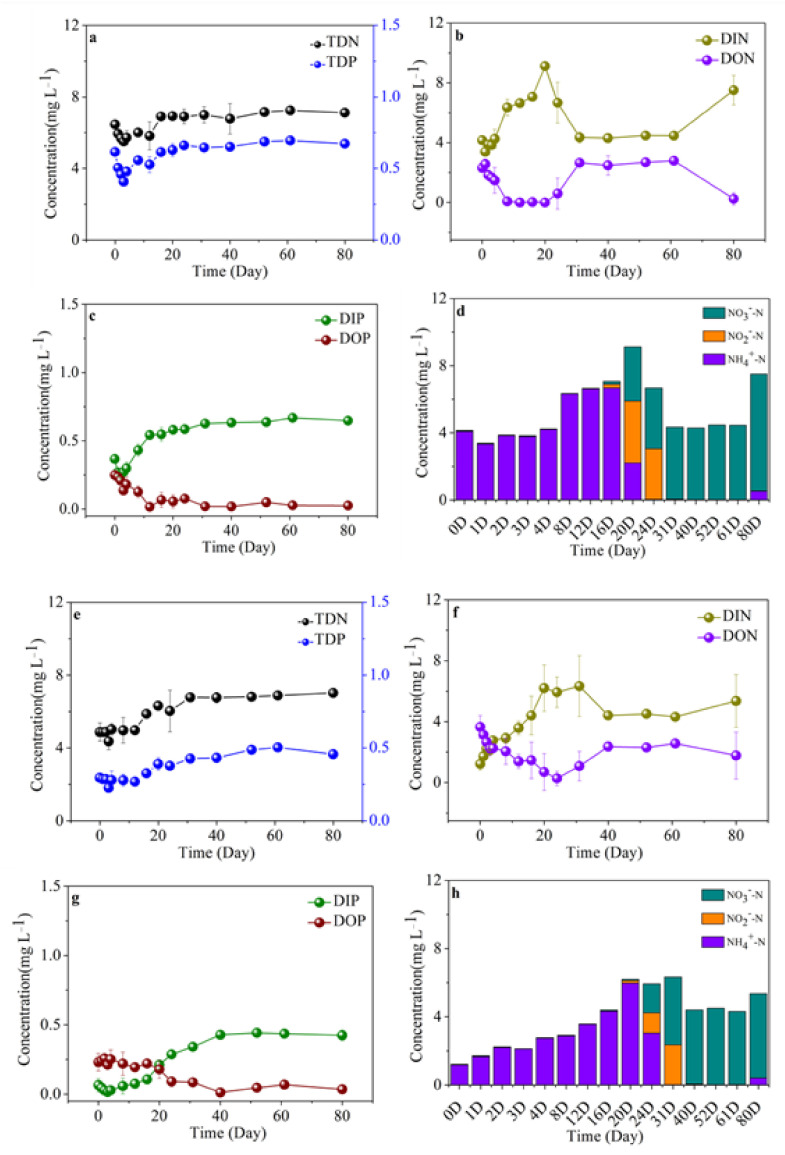
Variations in dissolved nitrogen and phosphorus concentrations and compositions during the 80-d degradation processes of the (**a**–**d**) algal-derived filtrate group and (**e**–**h**) algal residual exudative organic matter group. TDN—total dissolved nitrogen, TDP—total dissolved phosphorus, DIN—dissolved inorganic nitrogen, DON—dissolved organic nitrogen, DIP—dissolved inorganic phosphorus, DOP—dissolved organic phosphorus.

**Figure 10 ijerph-19-06981-f010:**
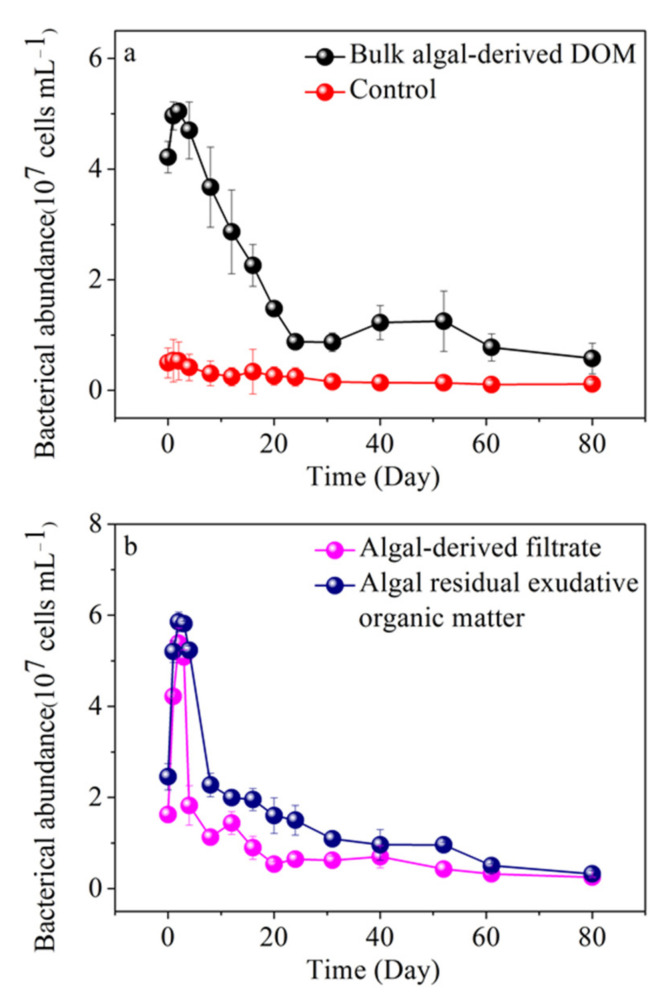
Dynamic changes in free-living bacterial abundance during 80-d degradation processes of (**a**) bulk algal-derived dissolved organic matter (DOM) group and in natural lake water group (as control), and (**b**) algal-derived filtrate group and algal residual exudative organic matter group.

**Table 1 ijerph-19-06981-t001:** Physicochemical properties of bulk algal-derived dissolved organic matter (DOM), algal-derived filtrate, and algal residual exudative organic matter. ND—not determined. DOC—dissolved organic carbon; DON—dissolved organic nitrogen; DOP—dissolved organic phosphorus.

	Parameter	Concentration (Mean ± Deviation)	Parameter	Concentration (Mean ± Deviation)
Bulk algal-derived DOM	DOC (mg L^−1^)	4.15 ± 0.56	C1 (R.U.)	0.78 ± 0.13
	DON (mg L^−1^)	1.63 ± 0.17	C2 (R.U.)	0.03 ± 0.01
	DOP (mg L^−1^)	0.40 ± 0.03	C3 (R.U.)	ND
	a (355) (m^−1^)	21.93 ± 1.80	C4 (R.U.)	ND
Algal-derived filtrate	DOC (mg L^−1^)	6.61 ± 0.73	C1 (R.U.)	0.43 ± 0.04
	DON (mg L^−1^)	1.66 ± 0.13	C2 (R.U.)	0.22 ± 0.02
	DOP (mg L^−1^)	0.22 ± 0.01	C3 (R.U.)	0.04 ± 0.04
	a (355) (m^−1^)	17.38 ± 0.16	C4 (R.U.)	0.03 ± 0.03
Algal residual exudative organic matter	DOC (mg L^−1^)	4.48 ± 0.58	C1 (R.U.)	0.87 ± 0.03
	DON (mg L^−1^)	3.00 ± 0.86	C2 (R.U.)	0.02 ± 0.03
	DOP (mg L^−1^)	0.20 ± 0.07	C3 (R.U.)	0.18 ± 0.10
	a (355) (m^−1^)	13.91 ± 0.29	C3 (R.U.)	0.01 ± 0.04

## Data Availability

Not applicable.

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
