# Peer review of "New Insights into Microbial Degradation of Cyanobacterial Organic Matter Using a Fractionation Procedure"

_ijerph, 2022, doi:10.3390/ijerph19126981_

Round 1

Reviewer 1 Report

This manuscript discussed microbial-regulated transformation of DOM from high-molecular-weight labile to low-molecular-weight recalcitrant. The topic is of interest for this journal, publication of this work can be considered. One comment is that the wind induced hydrodynamic effects in Lake Taihu may further affect the cyanobacteria cyanobacterial bloom. The authors need further clarification on this.

Reviewer 2 Report

1.      The Introduction section is in my opinion a bit too long and describes in too general terms the degradation of the organic substance. Therefore, I would recommend the authors describe the aspects related to the degradation of biomass produced by cyanobacteria and, in particular, to Mycrocysis species.

2.      The Materials and Methods section should be reduced as an extension and only those elements necessary for the reproducibility of the experiments should be retained.

Since Lake Taihu is a source of drinking water, the authors should question, at least in theory, the potential for toxins arising from the massive development of this species

Reviewer 3 Report

The authors have carried out important research necessary to prevent problems associated with cyanobacterial blooms caused by phytoplankton Microcystis in Lake Taihuthe and collection of clean drinking water. These studies provide new insights into microbial degradation of cyanobacterial organic matter using a fractionation procedure.

In general, the work makes a good impression. I found only small drawbacks. The text is not aligned in pages 10-11. The term "bacterial abundance" is preferred to "bacteraCal abundance" (the latter is used in Figure 10). It is desirable that the tables be on one page (do not break between two pages). Also, I'd like to recommend the authors to carefully read the text and correct some minor shortcomings. In my opinion, after minor revision this paper deserves to be published.
